# Formulation, Characterization and Permeability Studies of Fenugreek (*Trigonella foenum-graecum*) Containing Self-Emulsifying Drug Delivery System (SEDDS)

**DOI:** 10.3390/molecules27092846

**Published:** 2022-04-29

**Authors:** Dávid Sinka, Enikő Doma, Nóra Szendi, Jázmin Páll, Dóra Kósa, Ágota Pető, Pálma Fehér, Zoltán Ujhelyi, Ferenc Fenyvesi, Judit Váradi, Miklós Vecsernyés, Zsolt Szűcs, Sándor Gonda, Zoltán Cziáky, Attila Kiss-Szikszai, Gábor Vasas, Ildikó Bácskay

**Affiliations:** 1Department of Pharmaceutical Technology, Faculty of Pharmacy, University of Debrecen, Nagyerdei Körút 98, 4032 Debrecen, Hungary; sinka.david@pharm.unideb.hu (D.S.); doma.enike@gmail.com (E.D.); sz3ndi@gmail.com (N.S.); jazminpall@gmail.com (J.P.); kosa.dora@pharm.unideb.hu (D.K.); peto.agota@pharm.unideb.hu (Á.P.); feher.palma@pharm.unideb.hu (P.F.); ujhelyi.zoltan@pharm.unideb.hu (Z.U.); fenyvesi.ferenc@pharm.unideb.hu (F.F.); varadi.judit@pharm.unideb.hu (J.V.); vecsernyes.miklos@pharm.unideb.hu (M.V.); 2Doctoral School of Pharmaceutical Sciences, University of Debrecen, Nagyerdei Körút 98, 4032 Debrecen, Hungary; 3Institute of Healthcare Industry, University of Debrecen, Nagyerdei Körút 98, 4032 Debrecen, Hungary; 4Department of Pharmacognosy, University of Debrecen, Nagyerdei Körút 98, 4032 Debrecen, Hungary; szucs.zsolt@science.unideb.hu (Z.S.); gonda.sandor@science.unideb.hu (S.G.); vasas.gabor@science.unideb.hu (G.V.); 5Agricultural and Molecular Research and Service Institute, University of Nyíregyháza, Sóstói út 31/b, 4400 Nyíregyháza, Hungary; cziaky.zoltan@nye.hu; 6Department of Organic Chemistry, University of Debrecen, Egyetem tér 1, 4010 Debrecen, Hungary; kiss.attila@science.unideb.hu

**Keywords:** fenugreek, trigonella foenum-graecum, SEDDS, permeability assay, cytotoxicity, dissolution test

## Abstract

Fenugreek is used as a spice and a traditional herbal medicine for a variety of purposes, given its antidiabetic and antioxidant effects. Self-emulsifying drug delivery systems (SEDDS) of herbal drugs are targets of extensive research aiming to increase bioavailability and stability. The study’s objective was to formulate SEDDS containing Trigonella foenum-graecum extract to improve the stability of herbal extract and to increase their permeability through a Caco-2 monolayer. A characterized fenugreek dry extract was used for the formulations, while the SEDDS properties were examined by particle size analysis and zeta potential measurements. Permeability assays were carried out on Caco-2 cell monolayers, the integrity of which was monitored by follow-up trans-epithelial electric resistance measurements (TEER). Cytocompatibility was tested by the MTT method, and an indirect dissolution test was performed, using DPPH antioxidant reagent. Two different SEDDS compositions were formulated from a standardized fenugreek dry extract at either the micro- or the nanoemulsion scale with sufficient stability, enhanced bioavailability of the compounds, and sustained release from HPMC capsules. Based on our results, a modern, non-toxic, cytocompatible fenugreek SEDDS formulation with high antioxidant capacity was developed in order to improve the permeability and bioavailability of all components.

## 1. Introduction

Fenugreek (*Trigonella foenum-graecum* L.) is a herb cultivated on the eastern shores of the Mediterranean, Morocco, Egypt, and India. It has strong flavor and aroma, and the seeds are widely consumed in oriental cuisine as spices in food preparations, as well as in traditional medicine for a variety of purposes [1]. Several pharmacological effects of Trigonella have been researched, such as carminative, gastric stimulant, antioxidant, anti-inflammatory, anticarcinogenic and hepatoprotective effects [2]. The antidiabetic effects of fenugreek have also been studied. Both the seed and its methanolic, ethanolic and aquaeous extracts demonstrated hypoglycaemic properties through the inhibition of carbohydrate-digesting enzymes, and reductions in blood glucose level [3]. Its immunmodulatory effects through the stimulation of macrophages have also been described [4].

Fenugreek is also widely used for different purposes in pharmaceutical sciences. Its hydrogel-forming fibers make it a good choice for oral formulations in the field of bioavailability enhancement and drug release modification, even in cases of other herbal drug products [5]. Using fenugreek components as penetration enhancers for novel drug formulation are in the scope of the latest studies [6]. Polymers are universally used materials in the pharmaceutical field from formulation to packaging, with the focus on recycled, plant-based, or biodegradable options [7,8].

Self-emulsifying drug delivery systems (SEDDS) are mixtures of oils, surfactants and occasionally co-solvents, ideally of an isotropic nature. After oral administration, with gentle stirring in the gastrointestinal fluids, they form micro- or nanoemulsions containing the active pharmaceutical ingredient (APIs) [9]. SEDDS are easily manufactured and physically stable formulations that can improve the dissolution and absorption of lipophilic drugs and drug compounds, enhancing bioavailability [10]. The formulation of antioxidant agents into SEDDS can lead to increased oral bioavailability and enhanced efficacy [11], and result in better patient compliance [12].

The main differences between micro- and nanoemulsion systems are droplet size and stability. Nanoemulsion droplets are below 100 nm, while microemulsion particle sizes are between 200 and 300 nm. Microemulsions are thermodynamically stable and can be formed spontaneously, while nanoemulsions are rather kinetically stable and need external energy to form [13]. Nevertheless, the terminology of self-microemulsion systems and self-nanoemulsion systems (SMEDDS/SNEEDS) is still debatable [14].

SM/NEDDS formulations of herbal drugs are targets of extensive research. Several herbal drugs are poorly soluble, and have lipophilic properties, which leads to poor absorption and distribution, and decreased bioavailability and treatment efficacy. The application of SEDDS formulations of herbal medicines not only increases the solubility, and therefore the bioavailability, of the active ingredients, but also improves the stability and the dissolution profile, making controlled or sustained release achievable, and helping avoid the occasional irritation of the gastrointestinal tract [15,16]. The technology has shown impressive results in cases of silymarin (class IV of BCS, low solubility and low permeability) [17] and the poorly soluble antitumor agent indirubin [18]. In some cases, the main limiting factor of formulations is the poor chemical stability of the herbal compounds due to hydrolysis, as in the case of *Plantago lanceolata* [19]

A wide variety of fenugreek seed products are available commercially. These are mostly granules or capsules of ground fenugreek seeds, although studies have shown that the use of dried fenugreek extract instead of direct plant materials is more effective and has less side effects [20]. For the incorporation into the diet of diabetes mellitus patients, Trigonella has been formulated into a vegetable cereal mix [21]. Studies can be found on Trigonella-loaded polymeric nanoparticles used for enhanced bioavailability [22], but in most cases, fenugreek seed components are used as excipients for modern drug delivery system formulations [23]. The Caco-2 permeability of Trigonella compounds ranges from poor to good [24], and can be improved by penetration enhancers such as β-cyclodextrins [25].

Caco-2 permeability tests are commonly used for modeling the intestinal absorption and bioavailability of orally administered drugs. Several studies have targeted the Caco-2 permeability of different herbal drugs, such as curcumin [26], silymarin [27], *Salvia officinalis* [28], *Vitex agnus-castus* [29], and *Rauwolfia serpentina* [30], among others, most of them showing poor bioavailability. Caco-2 cell culture models have also been used for SEDDS-related studies concerning the effects of their cellular tight junctions [31], their bioavailability-increasing properties [32], comparing their permeability enhancement with nanoparticles [33], and studying the intestinal peptide absorption [34].

The study’s objective was to formulate SEDDS containing Trigonella foenum-graecum extract to improve the stability of the herbal extract and to increase its permeability through a Caco-2 monolayer. Specific transport studies of all active fenugreek extract components in SEDDS have been performed; these mechanisms have not been described yet. MTT viability tests of Caco-2 cells have also been performed to certify the cytocompatibility of the prepared compositions. The DPPH Radical Scavenging Activity of fenugreek-SEDDS samples was measured during an in vitro dissolution study. The correlation of antioxidant capacity and the anti-inflammatory property was also determined with an in vitro DPPH assay and a Superoxide Dismutase (SOD) enzyme activity cell assay.

## 2. Results

### 2.1. Characterization of Fenugreek Extract

A targeted search for bioactive natural products in the fenugreek extract has led to the identification of several putative flavonoid glycosides and putative steroid saponins, along with the successful detection of trigonelline and 4-hydroxyisoleucine. These compounds were identified according to their MS/MS spectra using the data from [35,36,37,38], while trigonelline and 4-hydroxyisoleucine showed identical spectroscopic characters as the authentic standards. The variability n the plant’s chemical constituents is highlighted by the wide range of oredicted logP values covered: while 4-hydroxyisoleucine and trigonelline have a logP of 0.135 and −2.481, respectively, typical flavonoid glycosides fell in the range −0.359 to 1.28, while trigofoenosides ranged from −3.8 to 0.4, depending on the number of saccharide moieties attached. On the other hand, as steroid saponins can be amphiphilic due to their saponin structure, direct interpretations of the logP might be of limited value. In case of flavonoids and trigonelline, the dissociation of -OH and -COOH groups results in more polar forms at physiological pH, resulting in a substantially increased log D.

Possibly due to the traces of residual phosphate and its incompatibility with ESI-MS, relatively poor QC values were obtained for many compounds, resulting in their rejection from the dataset. The reliably detectable compounds include five steroid glycosides, as well as six flavonoid glycosides. The latter include C-glycosides and O-glycosides.

### 2.2. Dynamic Light Scattering

The 1.59 refractive index and 0.010 absorption values of polystyrene particles were used to calculate the particle size distribution by volume of compositions TFG + S1 and TFG + S2. The average particle diameter of TFG + S1 was 1855 nm, and was 65.5 nm in the case of TFG + S2 (Figure 1 and Figure 2, respectively). Based on the results of the particle size analysis, we can determine that our product TFG + S1 is a self-microemulsifying drug delivery system (SMEDDS), while TFG + S2 is a self-nanoemulsifying drug delivery system (SNEDDS), although the nomenclature is debatable, as mentioned above.

### 2.3. Zeta Potential Analysis

The average zeta potential of TFG + S1 was −71.3 mV, with a standard deviation of 11.8 mV. The average zeta potential of TFG + S2 was −38.5 mV, with a standard deviation of 7.47 mV (Table 1). The magnitude of zeta potential in mV points to a stable behavior, with 0 to 5 mV meaning rapid coagulation or flocculation, 10 to 30 meaning incipient instability, 30 to 40 meaning moderate stability, 40 to 60 good stability, and over 61 excellent stability [39]. The stability of the composition TFG + 1 was considered excellent, and that of the composition TFG + 2 was considered moderate, according to this categorization.

### 2.4. Antioxidant Capacity

The graph (Figure 3) displays the inhibition percentages of reactive oxidative stubs of the samples in different dilutions. The inhibition values were calculated using the formula:$$I\%=\frac{A_{0}-A_{S\;}}{A_{0}}\times 100$$
where *A*_0_ is the absorbance of the 0.06 mM DPPH reagent, and *A_S_* is the absorbance of the sample. Each data point represents the mean ± S.D. of three experiments.

### 2.5. Superoxide Dismutase (SOD) Assay

The in vitro SOD enzyme activity assay was performed on Caco-2 cells (Figure 4). The SOD activity of the positive control cells treated with ascorbic acid dilution was taken as 100%. The SOD activity of the cells treated with TFG + S1 and TFG + S2 was higher than the SOD activity of cells treated with native fenugreek extract TFG. The SEDDS compositions S1 and S2 also showed antioxidant effects that were mildly higher than those of the negative control PBS.

### 2.6. Permeability Assay

Permeability tests were carried out on cell monolayers presenting TEER values between 800 and 1000 Ω cm^2^. The concentration of the penetrated compounds over time is displayed on Figure 5, Figure 6 and Figure 7. S1, S2, and the negative control PBS showed zero values for all compounds. Significantly more compound permeated from the SEDDS formulations TFG + S1 and TFG + S2 than from the native fenugreek extract TFG. The permeability curves can be compared with Tukey’s HSD post-hoc test, while the curves marked with different letters differ significantly.

### 2.7. TEER Follow Up

The membrane function was monitored using follow-up TEER measurements. After the permeability assay, TEER values increased in the fresh medium, and after 24 h the values were above 90% of the baseline TEER, as is displayed in Figure 8. The samples added to the inserts loosened the tight junctions of the cell layer, but in the fresh medium, this was restored slowly. The TEER value after 24 h was close to the baseline, which points to the survival of the cells in the fenugreek compositions. The reduction in TEER caused by Triton-X was constant until the end of the measurement, because the cells did not survive.

### 2.8. Citotoxicity Assay

The results of the MTT assays were used to construct a graph of cell viability percentage against the extract dilutions (Figure 9). The result of the blank PBS treatment was considered as 100% viability. The Triton X-100 treatment used as positive control resulted in a 6.7% cell viability. According to the MTT test results, the Trigonella extract and the SEDDS compositions were also well tolerated by the Caco-2 cells, as all cell viability values were over 74%. A cell survival rate over 70% is required to consider a material non-toxic. In the case of TFG, we experienced cell viability values over 100%, which could have been caused by the high antioxidant content of the plant extract enhancing the reduction in MTT. The Trigonella extract alone showed the highest cell survival rate, while there was no significant difference between the cytotoxicity values of the different SEDDS compositions.

Each cell viability value represents the mean ± S.D. of six independent parallel wells. At each concentration, the means of the different compositions were compared with one-way ANOVA tests followed by Tukey’s multiple comparison tests, where all the three compositions were compared to each other (Table 2.)

### 2.9. Dissolution Test

The graphs display the inhibition percentages of reactive oxidative stubs of samples taken from the dissolution medium over time (Figure 10 and Figure 11). The inhibition values were calculated using the formula:$$I\%=\frac{A_{0}-A_{S\;}}{A_{0}}\times 100$$
where *A*_0_ is the absorbance of the 0.06 mM DPPH reagent, and *A_S_* is the absorbance of the sample. Each data point represents the mean ± S.D. of five experiments.

We also displayed the antioxidant capacity of the Trigonella extract diluted in 100 mL distilled water as the positive control, and the antioxidant capacities of the SEDDS components (as negative control).

The dissolution was sustained and steady in the case of both the compositions and the capsule material. The hard gelatin capsules started to disintegrate after 30 min. The HPMC capsules achieved dissolution after 60 min.

At each time point, the means of the different compositions were compared with a Kolmogorov–Smirnov test, showing no significant difference between the dissolution profiles (*p* < 0.05).

## 3. Discussion

Fenugreek is a herb with a wide range of therapeutic effects. In our study, we formulated two different SEDDSs from the dry extracts of fenugreek and characterized the components according to the submitted study of Gonda et al.

4-hydroxyisoleucine is a non-proteinogenic amino acid that is less common in herbs. Its effect on insulin resistance has been described, and is caused by stimulated insulin secretion in the presence of high blood glucose levels [40,41]. As such, fenugreek is considered an antidiabetic herb [42]. Absorbability is the main limiting step in the bioavailability of amino acids, and its enhancement is currently a topic of intense research, with several approaches explored, such as absorption enhancers, chemical modification, and liposomes [43,44].

Fenugreek contains different diosgenin glucosides, which liberate diosgenin from the GI tract. Aglicon has beneficial effects on the recovery of pancreatic beta-cells and the down-regulation of hepatic gluconeogenesis glucose export enzymes [41]. Diosgenin is a steroidal saponine with a low absolute oral bioavailability, although it can be increased with permeability enhancers, both in vitro Caco-2 cell models and in in vivo rat experiments [25].

Trigonelline is an alkaloid that has many beneficial effects on insulin resistance, including reducing glucose uptake from the GI tract and enhancing pancreatic beta-cell recovery [45]. The solubility of trigonelline is considered to be good [46], but the absorption rate is moderate, and its improvement by nanoformulations is currently the target of research [47].

Apigenin inhibits α-glucosidase activity and increases the secretion of insulin [48]. It is a flavonoid with poor oral bioavailability, which can be improved with nanoformulations [47], solid dispersions [49], and submicron emulsions [50].

Poor oral bioavailability can be considered as one of the biggest challenges in drug formulation. Low water solubility, low membrane permeation and presystemic degradation are the main causes of insufficient oral bioavailability [51,52]. The formulation and optimization of SEDDS can enhance the bioavailability of lipophilic, low-permeability and unstable molecules [53,54].

The novelty of our study is the formulation of two different SEDDS compositions from a standardized fenugreek dry extract, at either the micro- or the nanoemulsion scale, with sufficient stability and enhanced bioavailability of the compounds.

SEDDS have been proven to be a useful technology for solubility and bioavailability enhancement, with easier scalability, and are also known to reduce metabolism and circumnavigate the hepatic first pass effect [55,56]; however, they have their own disadvantages, such as the low drug loading capacity, drug leakage, stability issues and the limited number of choices as regards dosage forms [57].

Fenugreek components are mostly used as excipients in modern drug delivery system formulations, and their incorporation into SEDDS compositions has not been investigated before.

Our permeability test examined the in vitro cellular uptake of 13 fenugreek compounds, including five steroid saponins, six flavonoid glycosides, 4-hydroxyisoleucine and trigonelline. The permeability of the two SEDDS compositions was significantly higher in the case of every compound, but obvious differences between the SEDDS compositions have not been found. Similar results of improved oral bioavailability were described after in vitro and in vivo permeability studies of SEDDS formulated from the herbal drugs phillygenin and tectorigenin [58,59]. The membrane integrity after the permeability test was monitored by follow-up TEER measurements.

The SEDDS formulations were characterized by zeta potential and particle size distribution studies. According to the results, we achieved a SMEDDS composition with excellent stability, and a SNEDDS composition with moderate stability, although categorizing self-emulsifying drug delivery systems based upon their size is debatable [14].

Former studies found that the antioxidant capacity of herbal components can be increased with penetration enhancement [60]. According to our antioxidant capacity assays, the SEDDS formulations with antioxidant properties on the same level as native fenugreek extracts formed using chemical reagent DPPH had higher antioxidant effects in the SOD cell assay. This better permeability may enhance the effect on the cell enzyme activity.

The indirect dissolution test of the compositions of SEDDS formed from hard gelatin and HPMC capsules was carried out. The hard gelatin capsule ensured a steady and sustained release for 30 min, and started to disintegrate after that. The HPMC capsule elongated the release to more than 60 min. The dissolved drug was monitored indirectly by DPPH reagent, exploiting the antioxidant properties of the fenugreek components, as was described by Kalantari et al. [19]. Several publications have described the stable and improved release of herbal drug SEDDS formulations using different dosage forms [15,19,59].

Cytotoxicity was studied by the MTT method. According to the results, the herbal extract and the SEDDS compositions were also well tolerated by the Caco-2 cells. We derived cell viability values over 100%, which may have been caused by the high antioxidant content of the plant extract enhancing the reduction in MTT [61,62]. Biocompatibility is a crucial parameter of penetration enhancers and SEDDS formulations [63]. Several studies were carried out at our department on the effects of SEDDS surfactants and co-surfactants on permeability and cell viability [19,63,64], and the compositions in this work were defined in terms of safety. The ISO 10993-5:2009(E) standard for the biological evaluation of medical devices and tests for in vitro cytotoxicity claims that an MTT assay with a cell survival rate of over 70% is necessary to determine a test item as non-toxic. Our compositions all meet this requirement.

## 4. Materials and Methods

### 4.1. Extraction Process

Fenugreek seeds of pharmaceutical quality were ground in a Retsch GM 200 Grindomix at 10,000 rpm for 2 min. A total of 60.0 g of seeds was extracted with 2400 mL of boiling methanol under reflux for 60 min, filtered on Whatman filter paper, and evaporated to dryness in a rotary evaporator. This resulted in 13.38 g of dry extract. Thereafter, the dry extract was re-dissolved in 30.00 mL PBS (phosphate buffer saline) and sterile filtered on a 0.22 µm PES membrane. The sterile extract was stored at 4 °C before use and used within 48 h of preparation. The concentrated liquid extract in PBS contained 7.833 mg mL^−1^ trigonelline and 8.258 mg mL^−1^ 4-hydroxyisoleucine.

### 4.2. Self Emulsifying Drug Delivery Systems

Isopropyl myristate was purchased from Merck (Darmstadt, Germany). Transcutol HP, Labrasol, Kolliphor RH40 and Capryol 90 were obtained from Gattefossé (Saint-Priest, France).

We formulated 5 different compositions of the liquid extract, the SEDDS excipients and PBS, as displayed on Table 3.

The various amounts of isopropyl miristate, Transcutol HP, Labrasol, Kolliphor RH40 and Capryol 90 were mixed well in a plastic tube. Concentrated liquid fenugreek extract or PBS was added, then the mixture was mixed with gentle vortexing until we achieved a visually homogenous appearance. All the experiments were performed immediately after preparation.

### 4.3. Caco-2 Cell Line

Caco-2 cells were maintained in plastic cell culture flasks in culture medium (Dulbecco’s modified eagle’s medium, supplemented with 3,7 g/L NaHCO_3_, 10% (*v*/*v*) heat-inactivated fetal bovine serum (FBS), 1% (*v*/*v*) non-essential amino acids solution, 1% (*v*/*v*) l-glutamine, 100 IU/mL penicillin, and 100 IU/ml streptomycin) in a humified atmosphere of 5% CO_2_ at 37 °C. The permeability and cytotoxicity tests were performed using cells in passages 20–40.

### 4.4. Dynamic Light Scattering

DLS measurements were carried out using a Malvern Zetasizer Nano ZSP. Here, 1 mL of the sample SEDDS composition was diluted with 900 mL distilled water. The samples were allowed to equilibrate for 5 min at 25 °C before performing 5 measurements consisting of 10 individual runs each. The refractive index and absorption values of polystyrene latex were used.

### 4.5. Zeta Potential Analysis

The zeta potential analysis was carried out using a Malvern Zetasizer Nano ZSP. Here, 1 mL of the sample SEDDS composition was diluted with 900 mL distilled water. The samples were allowed to equilibrate for 5 min at 25 °C before performing 5 measurements consisting of 50–100 individual runs each.

### 4.6. Antioxidant Capacity

The antioxidant capacities of the compositions were determined by DPPH (2,2-Diphenyl-1-picrylhydrazyl) assay. Here, 1 mg/mL ascorbic acid solution was used as the positive control. A serial dilution in the range of 10-fold to 100-fold was prepared from the compositions and the positive control. Then, 1 mL of the samples was added to 2 mL 0.06 mM DPPH reagent (Merck KGaA, Darmstadt, Germany), and the absorbance was measured with a Shimadzu UV-1900 spectrophotometer at the 517 nm wavelength.

### 4.7. Superoxide Dismutase (SOD) Assay

The Caco-2 cells were seeded on 12-well plates at a density of 10^5^ cells/well and grown in a CO_2_ incubator at 37 °C for six days. The cell culture medium was removed, and 1 mL 100× dilution of each composition was added to the cells, then incubated at 37 °C for 1 h. A 100× dilution of 1 mg/mL ascorbic acid was used as the positive control, while PBS was used as the negative control. Cells were harvested using a rubber rod and centrifuged at 1000× *g* for 10 min at 4 °C. The cell pellet was then homogenized in 20 mM HEPES buffer (1 mM EGTA, 210 mM mannitol, and 70 mM sucrose/g tissue) and centrifuged at 10,000× *g* for 15 min at 4 °C and pH 7.2. The SOD activity of the supernatant was analyzed using assay kits from Cayman Chemicals according to the manufacturer’s instructions (Cat. 706002, Cayman, Ann Arbor, MI, USA). All experiments were performed in triplicate.

### 4.8. Permeability Assay

For the permeability assay, Caco-2 cells were seeded on 24-well polycarbonate filter inserts at 80,000 cells/insert. The culture medium was replaced with fresh every 3–4 days. Trans-epithelial electronic resistance was measured using a Millicell ERS-2 voltohmmeter with a chopstick electrode pair. All cell monolayers presented TEER values between 800 and 1000 Ω cm^2^.

The inserts were placed into the wells of a fresh plate. The permeability assay was commenced with the addition of 400 µL of the sample solution to the apical chambers of the inserts. A 50 µL aliquot was taken from the basal chamber containing PBS immediately, and at 15, 30, 60, 120 and 240 min. The samples taken from the basal chamber were replaced with PBS. During the permeability assay, the plate was placed in an incubator (37 °C, 35 rpm).

The sample solutions used for the permeability assay were PBS (as blank) and the five studied compositions. Each sample solution was tested on 4 inserts in parallel.

### 4.9. Analytical Method and Metabolomics Quality Control Procedure

Sample preparation. Accurately measured 50 µL aliquots of the basal layer were extracted three times with 1000 µL of hexane and subsequently evaporated to dryness in a Christ RVC 2-33 CDplus vacuum centrifuge. The pellet was vortexed with 100 µL 96% ethanol supplemented with 0.1% HCOOH. This procedure enabled 84.50% and 90.38% recoveries of trigonelline and 4-hydroxyisoleucine, respectively, while removing fats and preventing the dissolution of phosphate from PBS at the same time.

Instrumentation. LC-MS measurements were carried out on an UHPLC system (Dionex Ultimate 3000RS) coupled to a Thermo Q Exactive Orbitrap mass spectrometer (Thermo Fisher Scientific Inc., Waltham, MA, USA) with an electrospray ionization source (ESI), using a Kinetex Polar C_18_ 100 × 3 mm × 2.6 µm × 100 Å column. The column temperature was set to 25 °C. A solvent gradient from solvent A (water + 0.1% FA) and solvent B (MeCN + 0.1% FA) at a flow rate of 0.2 mL min^−1^ was used, as follows: 0–2 min, 0% B; 2–14 min, 0–100% B; 14–15 min, 100% B; 15–16 min, 100–0% B; 16–25 min, 0% B. A 1 µL aliquout was injected for all samples.

The Orbitrap was operated in full MS mode at the *m*/*z* range 125–1875 and FWHM resolution 35,000, with polarity switching enabled. The capillary temperature was 320 °C, the maximum injection time was 100 ms, and the sheath gas and aux flow rates were 32 and 7 arb, respectively. The spray voltage was 4.0 kV and 3.8 kV for the positive ion mode and negative ion mode, respectively.

Quality control samples. Quality control (QC) samples were prepared as a mixture of equal volumes of single replicates of all treatment–time pairs. After a blank and an initial pre-equilibration of 2 + 4 injections of QC samples with a serial dilution of QC samples in between, samples were injected using a randomized injection order in blocks, with a QC injected at the beginning of each 7-sample block. QC samples were used to assess the relative standard deviation and linearity after targeted evaluation of the samples [65,66].

Peak detection. Raw instrument files were processed using mzMine 2.53 [67] after conversion into the centroided mzXml format, by manually searching for flavonoid glycosides and furostanol saponins trigoneosides and trigofoenosides, as described in [35,36,37,38], and subsequent targeted peak detection with mzMine with the following parameters: an intensity tolerance of 50%, noise level 10, *m*/*z* tolerance of 5 ppm and retention time tolerance of 0.2 min. The peaks were placed into lists with the “join aligner” algorithm, using the following parameters: *m*/*z* tolerance = 5 ppm, retention time tolerance = 0.15 min, weights = 1 and 1. Authentic standard peaks from trigonelline and 4-hydroxyisoleucine were also evaluated using this approach.

Method performance. The method was found to be linear for trigonelline and 4-hydroxyisoleucine from 0.5 to 20 µg mL^−1^ (R^2^ 0.9966, 0.9947, respectively). The intraday precision values were below 2.09% and 12.06% for trigonelline and 4-hydroxyisoleucine in the 5–20 µg mL^−1^ range of the calibration curve, respectively.

Data processing. After targeted peak detection, QC samples were evaluated for the reproducibility of features. Because of the expected poor measurement quality of PBS, relatively tolerant thresholds of R^2^ > 0.8 and RSD < 0.3 were set for inclusion. R^2^ values were calculated from blank and serial dilutions of QCs, while RSD values were calculated from the QC samples at the beginning of each sample block in the injection sequence. The features that passed these tests in QCs were selected for further evaluation.

### 4.10. Citotoxicity Assay

The MTT method was used for the cytotoxicity assay to determine the viability of Caco-2 cells after the treatment with compositions TFG, TFG + S1 and TFG + S2. A blank PBS treatment was used as a negative control, and the Triton X-100 (Sigma-Aldrich, Budapest, Hungary) treatment as the positive control. The cells were seeded at a density of 10,000 cells per well on flat-bottom 96-well tissue culture plates, and were allowed to grow for 7 days in a humified atmosphere of 5% CO_2_ at 37 °C.

A serial dilution in the range of 100-fold to 1000-fold was prepared from the compositions. The cytotoxicity assay was initiated with the removal of the cell culture medium and the washing of the cell. The sample solutions were added for a 30 min incubation period, then were removed. A further 3 h of incubation in cell culture medium containing MTT (0.5 mg/mL) followed (Sigma-Aldrich, Budapest, Hungary). The purple formazan crystals produced by cellular enzyme activity were dissolved from the bottom of the plate in a 25:1 isopropanol:hydrochloric acid solution. The absorbance was measured with a FLUOstar OPTIMA Microplate Reader at 570 nm against a 690 nm reference.

### 4.11. Dissolution Test

A modified dissolution test was carried out with compositions TFG + S1 and TFG + S2. Hard gelatin and HPMC capsule shells (Capsugel, Inc., Morristown, NJ, USA) were filled with 1 mL of each composition. Three capsules were put in a medium of 100 mL distilled water at room temperature and stirred at 120 rpm. A 1 mL aliquot was taken from the medium at 10, 20 and 30 min in the case of the hard gelatin capsules, and at 15, 30, 45 and 60 min in the case of the HPMC capsules.

The samples were added to 2 mL 0.06 mM DPPH (2,2-Diphenyl-1-picrylhydrazyl) reagent, and the absorbance was measured with a Shimadzu UV-1900 spectrophotometer on the 517 nm wavelength.

### 4.12. Statystical Analysis and Graph Display

All data were analyzed using GraphPad Prism (version 9; GraphPad Software, Inc., La Jolla, CA, USA). The calculation of mean and standard deviation and of graph display was carried out using Microsoft Excel 2016.

## 5. Conclusions

Based on our results, a modern, non-toxic, cytocompatible fenugreek SEDDS formulation with high antioxidant capacity was developed in order to improve the permeability and bioavailability of all components.

## Figures and Tables

**Figure 1 molecules-27-02846-f001:**
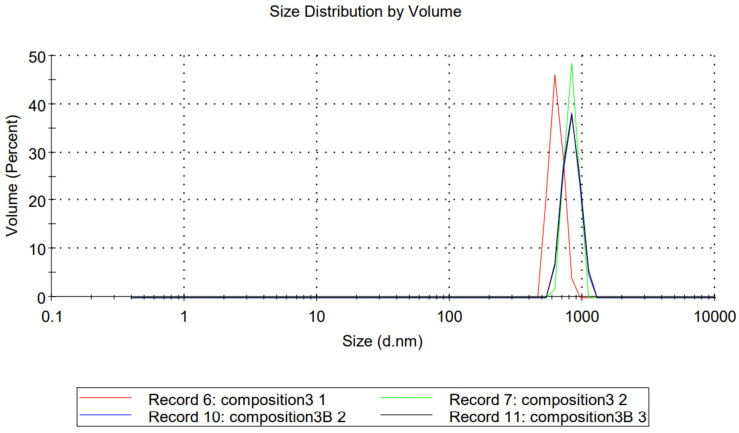
Size distribution by volume of TFG + S1. Each measurement consisted of 10 individual test runs. Average particle diameter was calculated by the software of the equipment.

**Figure 2 molecules-27-02846-f002:**
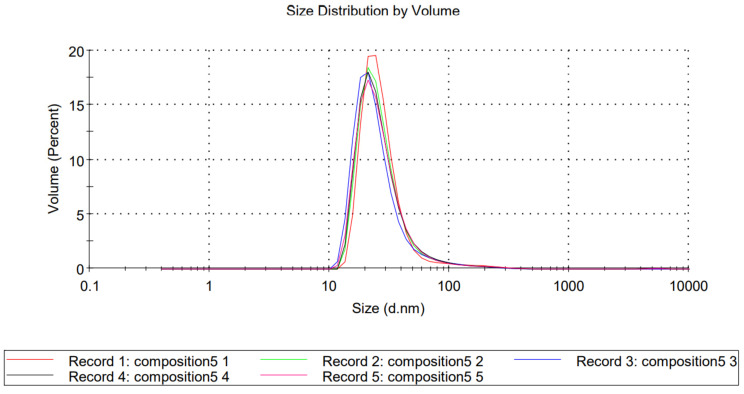
Size distribution by volume of TFG + S2. Each measurement consisted of 10 individual test runs. Average particle diameter was calculated by the software of the equipment.

**Figure 3 molecules-27-02846-f003:**
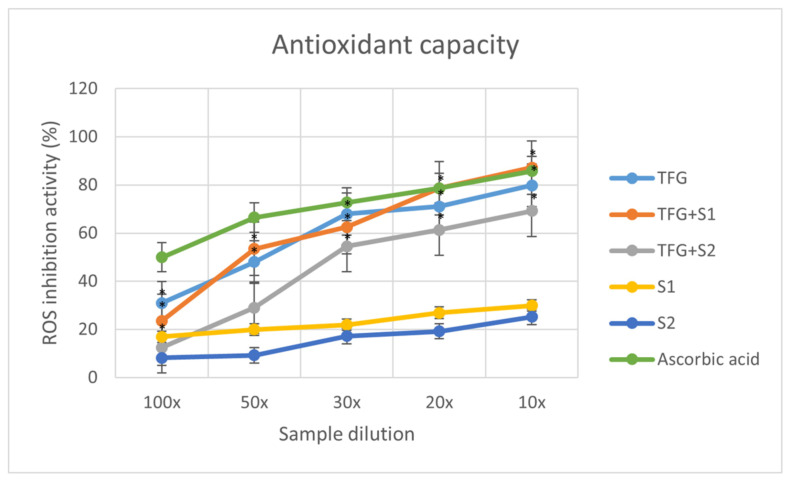
Antioxidant capacity of the 5 compositions and ascorbic acid, as a positive control. Each data point represents the mean ± S.D. of 3 experiments, with * *p* < 0.05 compared with the control.

**Figure 4 molecules-27-02846-f004:**
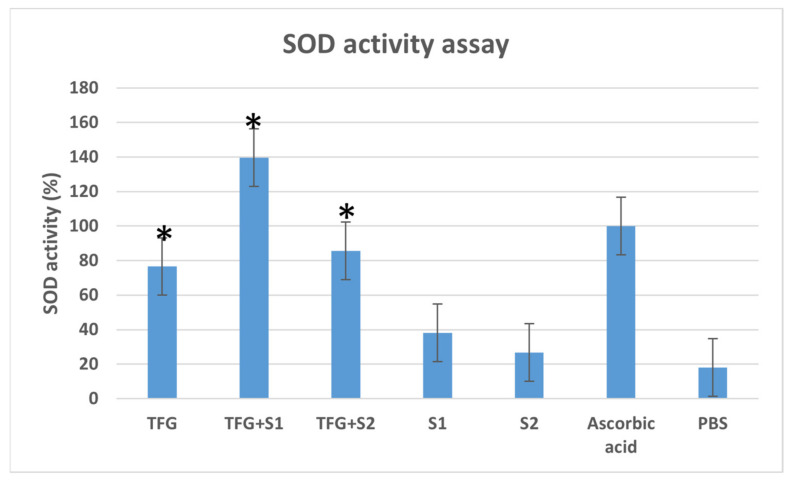
SOD enzyme activity in Caco-2 cells, expressed as the percentage of SOD activity in positive control cells, treated with ascorbic acid solution. Cells treated with PBS were used as the negative control. Data are expressed as the mean ± S.D. and n = 8, with * *p* < 0.05 compared with control.

**Figure 5 molecules-27-02846-f005:**
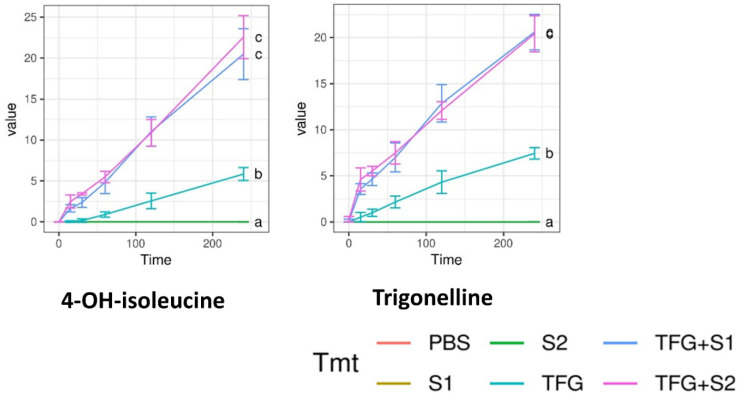
Abundance of Trigonella foenum-graecum compounds in the basal layer of Caco-2 cells, as measured by LC-ESI-MS. 4-hydroxyisoleucine and trigonelline were measured in positive ion mode. Data are given in µg·ml^−1^ for 4-hydroxyisoleucine and trigonelline. Abbreviations denote the compositions of the upper layer, as in Table 3. The curves marked with different letters differ significantly, according to Tukey’s HSD post-hoc test.

**Figure 6 molecules-27-02846-f006:**
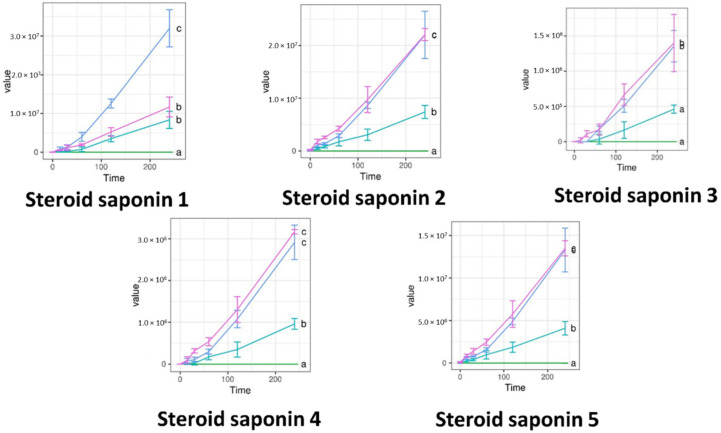
Abundance of Trigonella foenum-graecum compounds in the basal layer of Caco-2 cells, as measured by LC-ESI-MS. Steroid saponins were measured in positive ion mode. Data are given in raw abundance. Abbreviations denote compositions of the upper layer, as in Table 3. The curves marked with different letters differ significantly according to Tukey’s HSD post-hoc test.

**Figure 7 molecules-27-02846-f007:**
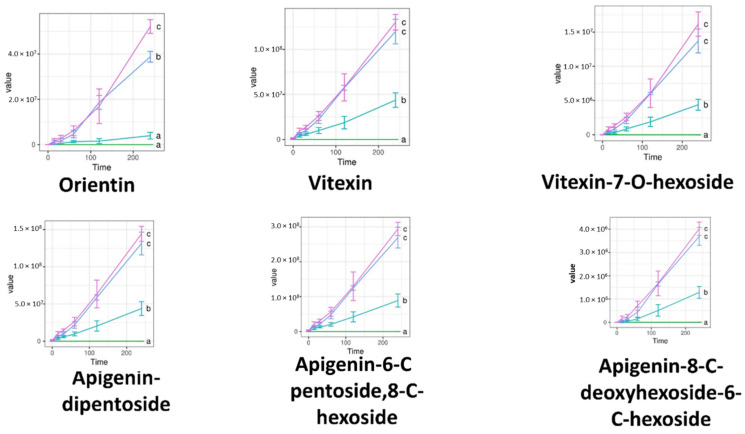
Abundance of Trigonella foenum-graecum compounds in the basal layer of Caco-2 cells, as measured by LC-ESI-MS. Flavonoid glycosides were measured in negative ion mode. Data are given in raw abundance. Abbreviations denote compositions of the upper layer, as in Table 3. The curves marked with different letters differ significantly according to Tukey’s HSD post-hoc test.

**Figure 8 molecules-27-02846-f008:**
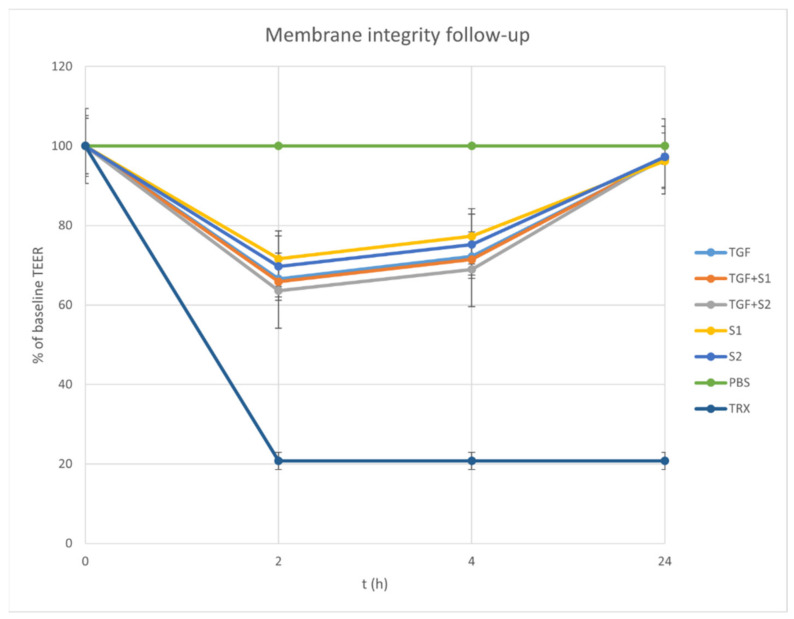
Trans-epithelial electrical resistance from the start of the permeability test to 24 h later. Triton-X was used as a positive control. The samples added to the inserts loosened the tight junctions of the cell layer, but in the fresh medium these were restored slowly. The TEER after 24 h was close to the baseline, which points to the survival of the cells in the fenugreek compositions. The reduction in TEER caused by Triton-X was constant until the end of the measurement, because the cells did not survive.

**Figure 9 molecules-27-02846-f009:**
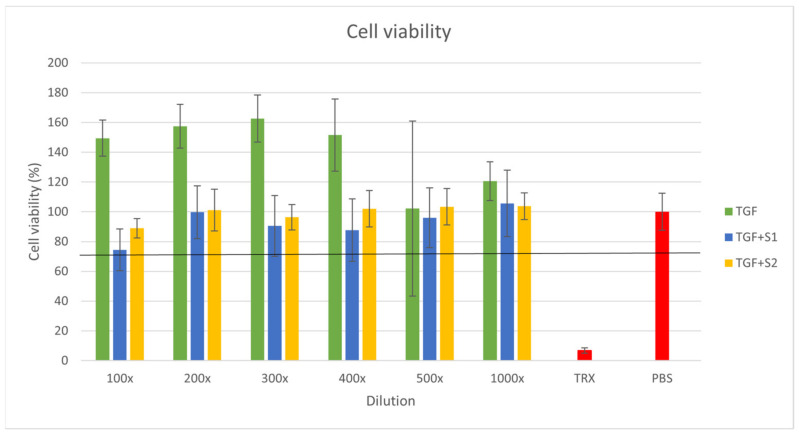
MTT cell viability assay results for Caco-2 cells. Cell viability expressed as the percentage of the negative control (PBS). Each data point represents the mean ± S.D. of 6 independent parallel wells. The cell viability was over 70% for every composition in each concentration. The Triton X-100 treatment used as positive control resulted in 6.7% cell viability. An ordinary one-way ANOVA test followed by a Tukey’s multiple comparison test was performed to compare the different compositions.

**Figure 10 molecules-27-02846-f010:**
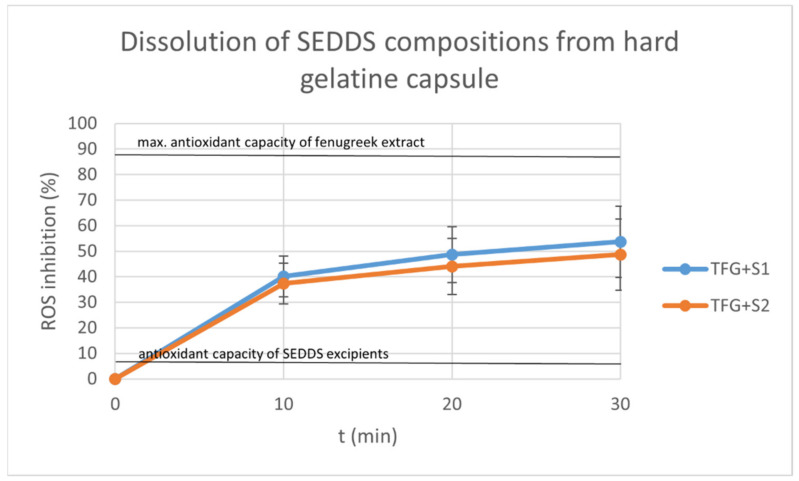
Dissolution of TFG + S1 and TFG + S2 from hard gelatin capsule over time. Each data point represents the mean ± S.D. inhibition percentage of 5 independent parallel dissolution test samples. The antioxidant capacity of Trigonella extract alone and the antioxidant capacity of the SEDDS components alone are also displayed. At each time point, the means of the different compositions were compared with a Kolmogorov–Smirnov test, and no significant difference was found between the dissolution profiles (*p* < 0.05).

**Figure 11 molecules-27-02846-f011:**
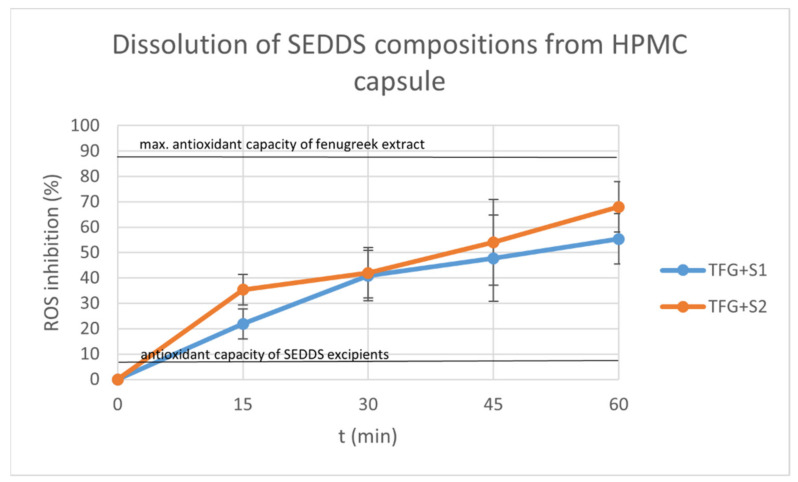
Dissolution of TFG + S1 and TFG + S2 from HPMC capsule over time. Each data point represents the mean ± S.D. inhibition percentage of 5 independent parallel dissolution test samples. The antioxidant capacity of the Trigonella extract alone and the antioxidant capacity of the SEDDS components alone are also displayed. At each time point, the means of the different compositions were compared with a Kolmogorov–Smirnov test, and no significant difference was found between the dissolution profiles (*p* < 0.05).

**Table 1 molecules-27-02846-t001:** Results of the zeta potential analysis of TFG + 1 and TFG + 2 with Malvern Zetasizer Nano ZSP. The stabilities of the compositions are considered excellent and moderate, respectively.

Formulation	Zeta Potential	Standard Deviation
TFG + S1	−71.3 mV	11.8 mV
TFG + S2	−38.5 mV	7.47 mV

**Table 2 molecules-27-02846-t002:** One-way ANOVA test followed by Tukey’s multiple comparison test performed on the Caco-2 cell viability results. ** *p* < 0.01; *** *p* < 0.001; **** *p* < 0.0001.

ANOVA Followed by Tukey’s Multiple Comparison Test Result	Level of Significance
TFG vs. TFG + S1, 10× dilution	****
TFG vs. TFG + S2, 100× dilution	***
TFG + S1 vs. TFG + S2, 100× dilution	****
TFG vs. TFG + S1, 200× dilution	**
TFG vs. TFG + S2, 200× dilution	ns
TFG + S1 vs. TFG + S2, 200× dilution	ns
TFG vs. TFG + S1, 300× dilution	**
TFG vs. TFG + S2, 300× dilution	ns
TFG + S1 vs. TFG + S2, 300× dilution	ns
TFG vs. TFG + S1, 400× dilution	**
TFG vs. TFG + S2, 400× dilution	ns
TFG + S1 vs. TFG + S2, 400× dilution	ns
TFG vs. TFG + S1, 500× dilution	ns
TFG vs. TFG + S2, 500× dilution	ns
TFG + S1 vs. TFG + S2, 500× dilution	ns
TFG vs. TFG + S1, 1000× dilution	ns
TFG vs. TFG + S2, 1000× dilution	ns

**Table 3 molecules-27-02846-t003:** Composition of the examined products.

	TFG	S1	TFG + S1	S2	TFG + S2
Fenugreek extract	6 mL	-	6 mL	-	6 mL
PBS	3 mL	6 mL	-	6 mL	-
isopropyl myristate	-	1 mL	1 mL	0.25 mL	0.25 mL
Transcutol HP	-	1 mL	1 mL	1.5 mL	1.5 mL
Labrasol	-	1 mL	1 mL	0.5 mL	0.5 mL
Kolliphor RH40	-	-	-	0.5 mL	0.5 mL
Capryol 90	-	-	-	0.25 mL	0.25 mL

## Data Availability

Data are available from the corresponding author with the permission of the head of the department. The data that support the findings of this study are available from the corresponding author (bacskay.ildiko@pharm.unideb.hu), upon reasonable request.

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
