# Peer review of "Formulation, Characterization and Permeability Studies of Fenugreek (Trigonella foenum-graecum) Containing Self-Emulsifying Drug Delivery System (SEDDS)"

_molecules, 2022, doi:10.3390/molecules27092846_

Round 1

Reviewer 1 Report

In this manuscript (molecules-1651925) the authors describe Formulation, characterization and permeability studies of fenugreek (Trigonella foenum-graecum) containing self-emulsifying drug delivery system (SEDDS). An enhancement in the introduction, results and discussions sections are needed before reaching on any decision.

  1. The introduction part of the manuscript is very poorly written need to be modified with latest/advanced materials. Latest references and work need to be incorporated in tabular form for a comparison with this work.
  2. Why the author chosen fenugreek?
  3. Elaborate the results of Zeta Potential Studies?
  4. English of the manuscript needs to polished and rectified thoroughly, with the changes in figures, legends and font size uniformly.
  5. To enhance the strength of the manuscript and a broader readership range, some important references, needs to be incorporated.

https://doi.org/10.3390/polym14020325; https://doi.org/10.1016/j.jff.2021.104405; Journal of Cleaner Production 248, 2020, 119251; Composites Part B: Engineering 183, 2020, 107655.

Author Response

Reply to the Reviewer 1.

Az űrlap teteje

Dear Reviewer 1.,

We would like to thank you all for your constructive comments in the review. Your comments provided valuable insights to refine its contents and analysis. In this document, we try to address the issues raised as best as possible. Our revision in the manuscript was highlighted by yellow colour.

  1. The introduction part of the manuscript is very poorly written need to be modified with latest/advanced materials. Latest references and work need to be incorporated in tabular form for a comparison with this work.

Thank you for your comment. We modified the introduction and completed it with references of recent studies.

  1. Why the author chosen fenugreek?

Our university research group studied fenugreek on more than one field. It has not been particularly characterized pharmacobotanically before, meanwhile the main focus of the Department of Pharmacology and Pharmacotherapy was diabetes and insulin resistance, and studying the pharmacological effects of fenugreek, among other herbal drugs (Szabó K, Gesztelyi R, Lampé N, Kiss R, Remenyik J, Pesti-Asbóth G, Priksz D, Szilvássy Z, Juhász B. Fenugreek (Trigonella Foenum-Graecum) Seed Flour and Diosgenin Preserve Endothelium-Dependent Arterial Relaxation in a Rat Model of Early-Stage Metabolic Syndrome. Int J Mol Sci. 2018 Mar 10;19(3):798. doi: 10.3390/ijms190307; Kiss, Rita, Szabó, Katalin, Gesztelyi, Rudolf, Somodi, Sándor, Kovács, Péter, Szabó, Zoltán, Németh, József, Priksz, Dániel, Kurucz, Andrea, Juhász, Béla, Szilvássy, Zoltán. Insulin-Sensitizer Effects of Fenugreek Seeds in Parallel with Changes in Plasma MCH Levels in Healthy Volunteers. International Journal of Molecular Sciences 2018 / 03 Vol. 19; Iss. 3. March, 2018. doi: 10.3390/ijms19030771). Our research group wanted to continue the work on fenugreek, with particular permeability studies and the stabilization using SEDDS. That was the reason why fenugreek was selected.

  1. Elaborate the results of Zeta Potential Studies?

Zeta potential measurements were carried out using a Malvern ZetaSizer Nano ZSP equipment. We displayed the values calculated by the software of the machine in a table. We also detailed the categorization of stability based on the zeta potential magnitude.

  1. English of the manuscript needs to polished and rectified thoroughly, with the changes in figures, legends and font size uniformly.

Thank you for your observation. We request English editing service for our manuscript before publishing, and changed font sizes according to the Molecules template.

  1. To enhance the strength of the manuscript and a broader readership range, some important references, needs to be incorporated.

Thank you for your suggestions, we incorporated all the four studies you mentioned.

Reviewer 2 Report

This manuscript aims to enhance the permeability of fenugreek via self-emulsifying drug delivery system. There are some major concerns that should be well addressed before reconsidering for publication in this journal.

  1. In the permeability study, the concentration rather than abundance of various compounds should be provided. Authors should use HPLC or LC-MS/MS to quantify the amount or concentration of permeated compounds. And authors should reorganize the Figure 4 to improve the readability.
  2. The description of Figure 5 is not detailed. Authors should provide the underlying mechanisms for the abrupt decrease and subsequent recovery of TEER for various SEDDS.
  3. In Figure 7 and Figure 8, the concentration of released fenugreek extract should be provided to replace ROS inhibition.
  4. In Figure 3, statistical analysis should be added.
  5. “Based on the results of the particle size analysis, our product TFG+S1 is a Self Microemulsifying Drug Delivery System (SMEDDS), while TFG+S2 is a Self Nanoemulsifying Drug Delivery System (SNEDDS)” is not consistent with the expression “nanoemulsion droplets are below 100 nm, while microemulsion particle size is between 200-300 nm”. Authors should be cautious with such expressions.

Author Response

Reply to the Reviewer 2.

Az űrlap teteje

Dear Reviewer 2.,

We would like to thank you all for your constructive comments in the review. Your comments provided valuable insights to refine its contents and analysis. In this document, we try to address the issues raised as best as possible. Our revision in the manuscript was highlighted by green colour.

  1. In the permeability study, the concentration rather than abundance of various compounds should be provided. Authors should use HPLC or LC-MS/MS to quantify the amount or concentration of permeated compounds. And authors should reorganize the Figure 4 to improve the readability.

As stated in the header of Fig.4., data are given in µg mL-1 for 4-hydroxyisoleucine and trigonelline, and raw abundance for the rest of the compounds.

In targeted metabolomic approaches, it is acceptable to provide raw abundances, if measurement quality is assessed (see 4.7. for details).  To make the presentation clearer, compounds that were quantified were kept in Fig.4. only, the rest (where abundance data are available) were presented in figure 5 and 6. Authentic standards for Trigonella steroid saponins are not available from vendors. Flavonoids are only minor contributors to the beneficial effects of the plant; therefore we did not aim their quantification.

  1. The description of Figure 5 is not detailed. Authors should provide the underlying mechanisms for the abrupt decrease and subsequent recovery of TEER for various SEDDS.

Thank you for your suggestions. We detailed the mechanism in the figure description and in the main text.

  1. In Figure 7 and Figure 8, the concentration of released fenugreek extract should be provided to replace ROS inhibition.

We measured the antioxidant capacity because fenugreek contains multiple components which lead to the antioxidant effect (Figure 9., 10). We did not want to choose one or two of the components to detect, because the effect of herbal drugs is always complex. In this way the results showed the effect of total released components. It may refer the in vitro biopharmaceutical process.

  1. In Figure 3, statistical analysis should be added.

We completed Figure 3 and the figure description with statistical analysis.

  1. “Based on the results of the particle size analysis, our product TFG+S1 is a Self Microemulsifying Drug Delivery System (SMEDDS), while TFG+S2 is a Self Nanoemulsifying Drug Delivery System (SNEDDS)” is not consistent with the expression “nanoemulsion droplets are below 100 nm, while microemulsion particle size is between 200-300 nm”. Authors should be cautious with such expressions.

Thank you for our observation. The nomenclature of SMEDDS and SNEDDS is strongly debatable, different authors describe different categorization. The categorization you cited missing the systems with particle size between 100 and 200 nms, and above 300 nm. We used the term SMEDDS to highlight that particle size is over 100 nm.

Reviewer 3 Report

The authors described the preparation of a SEDD system for fenugreek extract to increase bioavailability and stability. Using the SEDDS containing fenugreek, the authors tested the antioxidation, permeability and cytotoxicity. They concluded that the fenugreek SEDDS formulation shows high antioxidant capacity, permeability, and low cytotoxicity. However, this manuscript has a few flaws.

Major comment

  1. The quality of figures are poor and should be replotted.
  2. The detail in the preparation of fenugreek is missing and must be described.
  3. The antioxidation assay is better done in cells as well, since it is biased to correlate the antioxidant capacity with cytotoxicity because the antioxidation is done only using a biochemical method instead of a cell assay.
  4. S1 and S2 should be used to assay the cytotoxicity as the antioxidant and permeability assays.
  5. The protective effect of TGF only is more effective than TGF-S1 and TGF-S2 from the antioxidant assay (Fig.3) and cytotoxicity (Fig. 6). This leads to one question: though the permeability and stability are increased, what is the advantage of SEDD for?

Minor comment

Citotoxicity should be cytotoxicity in section 4.3

Author Response

Reply to the Reviewer 3.

Az űrlap teteje

Dear Reviewer 3.,

We would like to thank you all for your constructive comments in the review. Your comments provided valuable insights to refine its contents and analysis. In this document, we try to address the issues raised as best as possible. Our revision in the manuscript was highlighted by blue colour.

  1. The quality of figures are poor and should be replotted.

Thank you for your suggestion, we replotted all figures in higher resolution.

  1. The detail in the preparation of fenugreek is missing and must be described.

Homogenization and the extraction process was described in 4.1., while sample preparation for LC-MS was described in 4.7. We think these sections are detailed enough to enable reproduction of the experiments. If you do not agree, please explicitly specify what details are missing.

  1. The antioxidation assay is better done in cells as well, since it is biased to correlate the antioxidant capacity with cytotoxicity because the antioxidation is done only using a biochemical method instead of a cell assay.

We have not carried out antioxidant cell assay because antioxidant effects are not the main focus of this research, we wanted to highlight on the detailed permeability characterization of fenugreek seed extracts and its bioavailability study. We also wanted to prove that SEDDS formulations do not decrease the antioxidant capacity of the native fenugreek extract.

Nevertheless, we started to prepare the antioxidant assay suggested by reviewer, but we had only 10 days to reupload the article with the major revisions, which was not enough to conduct the antioxidant cell assay, but we started to grow the cells for it, and we will add it as a supplement, if it could enhance the strength of the manuscript.

  1. S1 and S2 should be used to assay the cytotoxicity as the antioxidant and permeability assays.

Our research group carried out several studies in the recent years focusing on penetration enhancers and SEDDS. All the materials we used as excipients in this research have been studied before in terms of cytotoxicity, and proven to be safe to use. (Ujhelyi, Z.; Fenyvesi, F.; Váradi, J.; Fehér, P.; Kiss, T.; Veszelka, S.; Deli, M.; Vecsernyés, M.; Bácskay, I. Evaluation of cytotoxicity of surfactants used in self-micro emulsifying drug delivery systems and their effects on paracellular transport in Caco-2 cell monolayer. Eur. J. Pharm. Sci. 2012, 47, 564–573.)

  1. The protective effect of TGF only is more effective than TGF-S1 and TGF-S2 from the antioxidant assay (Fig.3) and cytotoxicity (Fig. 6). This leads to one question: though the permeability and stability are increased, what is the advantage of SEDD for?

The aim of our study was to enhance permeability of the fenugreek components, which was successful based on the permeability assay. The SEDDS formulations provide better penetration for all studied components than the native fenugreek extract, while some of the components have poor bioavailability from the native extract. Considering the coating properties of some fenugreek polysaccharides in the gastrointestinal tract, the SEDDS formulations enhance the bioavailability of the fenugreek active ingredients. Improving the antioxidant and cytotoxic properties was not intended in this research.

Minor comment

Citotoxicity should be cytotoxicity in section 4.3

Thank you for your observation, we corrected the spelling.

Round 2

Reviewer 1 Report

I have checked the revised version of the manuscript, Authors replied all the comments satisfactorily, now it can be accepted for future process aftet minor spelling checks.

Author Response

Thank you very much for your comments and support during the revision process!

Reviewer 2 Report

Authors have addressed my concerns, I recommend its publication.

Author Response

Thank you very much for your positive reply! Thank you for your permanent support during revision process!

Reviewer 3 Report

The revised manuscript is not satisfied and can be further improved before acceptable for publication. 

  1. The preparation of fenugreek SEDDS is still described in the method.
  2. The antioxidant cell assay is not performed.
  3. Not every reader is familiar with the SEDD system. If the cytotoxicity and other assays of S1 and S2 have been  performed in previous studies, then the related results should be included in the results or discussion to make a clear profile for general readers.
  4. As the author's response, the aim of this study focus on the enhancement of permeability of fenugreek SEDD. However, the ultimate goal for the enhancement of the permeability of fenugreek is able to use the new formulation to improve antioxidant effect on medical use. From the present results, although the permeability is great improved, the antioxidant outcome receives no benefit from the increase of permeability. Then, the increase of permeability is useless, and what  is the meaning to make fenugreek SEDDs. We can directly use TGF !

Author Response

Reply to the Reviewer 3.

Az űrlap teteje

Dear Reviewer 3.,

We would like to thank you all for your constructive comments in the review. Your comments provided valuable insights to refine its contents and analysis. In this document, we try to address the issues raised as best as possible. Our revision in the manuscript was highlighted by red colour.

  1. The preparation of fenugreek SEDDS is still described in the method.

We detailed the preparation of SEDDS compositions in chapter 4.2.

  1. The antioxidant cell assay is not performed.

We performed a SOD enzyme activity assay on Caco-2 cells, and completed the results, methods and discussion parts.

  1. Not every reader is familiar with the SEDD system. If the cytotoxicity and other assays of S1 and S2 have been performed in previous studies, then the related results should be included in the results or discussion to make a clear profile for general readers.

Thank you for your suggestion. We completed the discussion with reference to the former studies related to this work.

  1. As the author's response, the aim of this study focus on the enhancement of permeability of fenugreek SEDD. However, the ultimate goal for the enhancement of the permeability of fenugreek is able to use the new formulation to improve antioxidant effect on medical use. From the present results, although the permeability is great improved, the antioxidant outcome receives no benefit from the increase of permeability. Then, the increase of permeability is useless, and what is the meaning to make fenugreek SEDDs. We can directly use TGF !

Thank you for your observation. We carried out cell antioxidant assay as you suggested, which confirmed that the higher permeability leads to better antioxidant activity in the cells, even if the antioxidant properties are on the same level in the chemical reagent assay.
